# Hospitalisation trends in India from serial cross-sectional nationwide surveys: 1995 to 2014

Anamika Pandey,[1,2] George B Ploubidis,[3] Lynda Clarke,[2] Lalit Dandona[1,4]

[1]Public Health Foundation of India, Gurugram, National Capital Region, India
[2]Department of Population Health, London School of Hygiene and Tropical Medicine, London, UK
[3]Centre for Longitudinal Studies, Department of Social Science, UCL- Institute of Education, University College London, UK
[4]Institute for Health Metrics and Evaluation, University of Washington, Seattle, Washington, USA

**Correspondence to**
Anamika Pandey;
anamika.pandey@phfi.org

## ABSTRACT

**Objectives** We report hospitalisation trends for different age groups across the states of India and for various disease groups, compare the hospitalisation trends among the older (aged 60 years or more) and the younger (aged under 60 years) population and quantify the factors that contribute to the change in hospitalisation rates of the older population over two decades.

**Design** Serial cross-sectional study.

**Setting** Nationally representative sample, India.

**Data sources** Three consecutive National Sample Surveys (NSS) on healthcare utilisation in 1995–1996, 2004 and 2014.

**Participants** Six hundred and thirty-three thousand four hundred and five individuals in NSS 1995–1996, 385 055 in NSS 2004 and 335 499 in NSS 2014.

**Methods** Descriptive statistics, multivariable analyses and a regression decomposition technique were used to attain the study objectives.

**Result** The annual hospitalisation rate per 1000 increased from 16.6 to 37.0 in India from 1995–1996 to 2014. The hospitalisation rate was about half in the less developed than the more developed states in 2014 (26.1 vs 48.6 per 1000). Poor people used more public than private hospitals; this differential was higher in the more developed (40.7% vs 22.9%) than the less developed (54.3% vs 40.1%) states in 2014. When compared with the younger population, the older population had a 3.6 times higher hospitalisation rate (109.9 vs 30.7) and a greater proportion of hospitalisation for non-communicable diseases (80.5% vs 56.7%) in 2014. Among the older population, hospitalisation rates were comparatively lower for females, poor and rural residents. Propensity change contributed to 86.5% of the increase in hospitalisation among the older population and compositional change contributed 9.3%.

**Conclusion** The older population in India has a much higher hospitalisation rate and has continuing greater socioeconomic differentials in hospitalisation rates. Specific policy focus on the requirements of the older population for hospital care in India is needed in light of the anticipated increase in their proportion in the population.

## Strengths and limitations of this study

► The use of large-scale data from nationwide surveys in India over two decades provides the most updated trends for hospitalisation.
► The evidence on the changing hospitalisation rate by age groups and the reasons behind the increased hospitalisation of the older population is timely for policy formulation given the population ageing and shifting disease burden.
► It was not possible for us to study the contribution of the supply side factors in the increased hospitalisation.
► Self-reported data and the nature of cross-sectional data may lead to recall and reporting biases, which may have affected the accuracy of the results.

older population in India suffer from a higher burden of disease at older ages, particularly chronic diseases and disabilities.[3–11] The ageing population in India will continue to be one of the major determinants of the change in disease burden over the next two decades.[5] Higher disease burden rates at older ages result in greater demand for healthcare, particularly hospitalisation.[12–15] Hospital care is an important aspect of any health system, especially regarding the treatment of the more vulnerable older segment of the population.[16 17]

Monitoring change in hospitalisation rates is important to highlight the necessity for health policies to allocate resources and services to respond to the diverse healthcare needs of different segments of the population. Studies in India have analysed hospitalisation, but they are restricted in their approach and lack comprehensive assessment of rate over time.[16 18–22] The purpose of this study was to analyse hospitalisation trends from nationally representative data between 1995 and 2014 for different age groups across the less and more developed states of India, and for various disease groups. In addition to this, we aimed to compare the hospitalisation trends of the older population with the population

## INTRODUCTION

The improvement in life expectancy in India has not been matched by the improvements in levels of health of the population.[1 2] The

under 60 years, and quantify the propensity and compositional change that may contribute to the change in hospitalisation rates of the older population.

## METHODS

### Ethics statement

The study is based on secondary data from the National Sample Survey (NSS) with no identifiable information on the survey participants. Exemption from ethics approval for analysis of the NSS data was obtained from the institutional ethics committees of the Public Health Foundation of India and the London School of Hygiene and Tropical Medicine.

### Data sources and participants

We used individual-level data from the NSS on healthcare utilisation conducted in all Indian states in 1995–1996, 2004 and 2014.[23–25] These surveys record the utilisation of healthcare for both inpatient and outpatient care, with hospitalisation episodes in 365 days reference period recorded in detail. In addition, information of certain aspects of the condition of the older population was also collected. Individual-level data were collected for a nationally representative sample of 633 405 in NSS 1995–1996, 385 055 in NSS 2004 and 335 499 in NSS 2014. The sample of the older population in these surveys was: 35 274 in NSS 1995–1996, 35 567 in NSS 2004 and 28 397 in NSS 2014. Samples with missing values for the independent variables were dropped, meaning that we did a complete case analysis. The proportion of missing cases on any independent variable across the three surveys was <4% of the total sample (see online supplementary table 1). Although there was variation in sample size, the sample design was uniform across the three surveys. This permits the construction of comparable variables which could be used to make statistical inferences about change in parameter estimates.

Initial analyses of trends and differentials in hospitalisation rates were performed on all persons surveyed including deceased members. However, for the subsequent descriptive, multivariable and decomposition analyses performed on the older population, the deceased was excluded because the questions on several important background variables were only asked to the older persons who were alive on the date of survey. The sample of deceased older population is reported in online supplementary table 1.

### Measures

Our outcome variable was hospitalisation rate defined as the number of episodes of hospitalisation in 365 days reference period per 1000 of the population exposed to the risk. The cause of hospitalisation was categorised into non-communicable diseases and injuries (NCDs) and communicable diseases and nutritional disorders (CDs) using the Global Burden of Disease 2013 classification.[2]

The diseases included in the two broad categories are listed in online supplementary table 2.

We used monthly per capita consumption expenditure (MPCE) adjusted to the household size and composition as a proxy for economic status. The equivalence scale used was $e_h = (A_h + 0.5K_h)^{0.75}$, where $A_h$ was the number of adults in the household and $K_h$ was the number of children aged 0–14 years. Parameters were set on the basis of estimates summarised by Deaton.[26] The state-specific adult equivalent mean MPCE was used as a cut-off to categorise households into poor and non-poor.

We present analyses at the state level for the 29 states and 7 union territories in India by classifying them into two groups—less developed and more developed states. The less developed states include the 18 states namely, 8 empowered action group states (Bihar, Jharkhand, Madhya Pradesh, Chhattisgarh, Uttar Pradesh, Uttaranchal, Odisha and Rajasthan), 8 north-eastern states (Assam, Arunachal Pradesh, Manipur, Mizoram, Meghalaya, Nagaland, Sikkim and Tripura), Himachal Pradesh and Jammu and Kashmir.[27] State-specific rates were estimated for the 19 major states of India, with a population over 10 million in 2011 census, accounting for 97% of India's population. For comparison, Bihar, Madhya Pradesh, Uttar Pradesh and Andhra Pradesh were considered as undivided states at all survey points.

The Andersen's model of healthcare utilisation was used to study the association of individuals' predisposing, enabling and need variables with hospitalisation.[28] Based on the availability of data age, sex, marital status, caste[i] and education were identified as predisposing variables; place of residence, states, economic independence, economic status and living arrangement as enabling factors and physical mobility status, current self-rated health (SRH) and SRH compared with previous year as the need variables, which are likely to affect hospitalisation in the older population. These variables were dichotomised for all analyses.

### Statistical methods

Descriptive analyses were used to examine the change in hospitalisation rate for all diseases, NCDs and CDs at both aggregate and subgroup levels for all ages, and the change in the composition of the older population in India between 1995 and 2014.

A logit model was used to evaluate the effect of covariates on the probability of hospitalisation in the older population. The model employed was of the form:

$$Ln[P_i / (1 - P_i)] = \sum \beta_i X_i \qquad (1)$$

where $Ln[P_i / (1 - P_i)]$ was the log odds of hospitalisation, $X_i$ was a vector of explanatory variables and $\beta_i$ was a vector of regression coefficients. The model was checked

---

[i]Caste in India is a social stratification of communities into four groups, namely scheduled castes (SCs), scheduled tribes (STs), other backward castes, and other castes. SC/STs are officially designated disadvantaged groups in India.

for multicollinearity. Fit of the model was assessed using the P value of the F-adjusted mean residual goodness-of-fit statistic. A P value <0.05 was not considered a good fit.

A regression decomposition technique was used to decompose the change in hospitalisation rate into its constituent parts.[29–31] A multivariable logit model was estimated for each period. For example, the equation for the period 1995–1996 was:

$$Ln[P_i/(1-P_i)]_{(1995-1996)} = \beta_0 + \beta_i X_{i(1995-1996)} + \cdots + \\ \beta_n X_{n(1995-1996)} \\ i = 1,2,3,4\ldots n \quad (2)$$

while the equation for the period 2014 was

$$Ln[P_i/(1-P_i)]_{(2014)} = \beta_0 + \beta_i X_{i(2014)} + \ldots + \\ \beta_n X_{n(2014)} \\ i = 1,2,3,4\ldots n \quad (3)$$

The difference $Ln[P_i/(1-P_i)]_{(2014)} - Ln[P_i/(1-P_i)]_{(1995-1996)}$ was decomposed using equation (4), which considered 1995–1996 as the base period.

$$Logit_{(2014)} - Logit_{(1995-1996)} = [(\beta_{0(2014)} - \beta_{0(1995-1996)}) + \\ \sum P_{ij(1995-1996)}(\beta_{ij(2014)} - \beta_{ij(1995-1996)})] + \\ \sum \beta_{ij(1995-1996)}(P_{ij(2014)} - P_{ij(1995-1996)}) + \ldots + \\ \sum(\beta_{ij(2014)} - \beta_{ij(1995-1996)}) \\ (P_{ij(2014)} - P_{ij(1995-1996)}) \quad (4)$$

Where,

$P_{ij(2014)}$=proportion of $j$th category of the $i$th covariate in NSS 2014.

$P_{ij(1995-1996)}$=proportion of $j$th category of the $i$th covariate in NSS 1995–1996.

$\beta_{ij(2014)}$=coefficient for the $j$th category of the $i$th covariate in NSS 2014.

$\beta_{ij(1995-1996)}$=coefficient for the $j$th category of the $i$th covariate in NSS 1995–1996.

$\beta_{0(2014)}$=regression constant in NSS 2014.

$\beta_{0(1995-1996)}$=regression constant in NSS 1995–1996.

This procedure yields three components: (1) propensity defined as the change brought by variation in the impact of determinants; (2) composition defined as the change due to variation in the proportion of determinants and (3) interaction which reflects the change as a result of the interplay between compositional and propensity change.[32] We used P values for the Wald test to assess the difference between the coefficients from the two logit models. The estimates were generated using survey sampling weights, and the survey design features including the cluster design effect were taken into account to calculate the 95% confidence intervals (CI). This was done using the 'svyset' command in STATA V.13.1 (StataCorp, College Station, Texas, USA).

The study design, analysis and reporting were conducted in accordance with the STROBE guidelines for reporting observational studies as shown in online supplementary file 1.

## RESULTS

### Hospitalisation trends and differentials

The annual hospitalisation rate per 1000 increased 2.23 times between 1995 and 2014; the increase was higher for NCDs than CDs (3.61 vs 2.25 times) (table 1). The contribution of NCDs to total hospitalisation increased from 38.6% in 1995–1996 to 62.2% in 2014. The hospitalisation rate increased with age, and was highest for the population aged 70 years or more. The hospitalisation rate increased 2.21 times for older population, and 2.01 times for population under 60 years between 1995 and 2014. When compared with younger population, the older population had more than three times higher hospitalisation rates, and a greater proportion of hospitalisations for NCDs.

Males and females under 60 years had similar hospitalisation rates, while the older males had 64% higher hospitalisation rate than the older females in 1995–1996 (figure 1). The gender gap reduced for the older population by 2014 because of the higher increase in hospitalisation rate for the females compared with the males (2.71 vs 1.89 times). As compared with poor, among older population, the non-poor had 62% higher hospitalisation rate, while among population under 60 years, the non-poor had 36% higher hospitalisation rate in 2014. In 1995–1996, the urban residents aged 60 years or more had 71% higher hospitalisation rate than the rural residents, which declined to 34% higher in 2014. As compared with the less developed states, the hospitalisation rate in the more developed states was 2.82 times higher for the older population and 2.07 times higher for those under 60 years; however, the differential became similar by 2014.

The more developed states had 2.21 and 1.86 times higher hospitalisation rate than the less developed states in 1995–1996 and 2014, respectively (table 2). Between 1995 and 2014, the increase in hospitalisation rate was higher in the less developed states compared with the more developed states, more so for the older population for all diseases (3.12 vs 1.89 times), NCDs (4.50 vs 2.63 times) and CDs (2.59 vs 1.66 times). The hospitalisation rate for older population by disease groups in the major states of India is shown for 1995–1996, 2004 and 2014 in online supplementary table 3.

Between 1995 and 2014, the hospitalisation in public hospitals declined from 44.9% to 38.4% (table 3). The use of public hospitals was higher in the less developed than the more developed states in 2014 (47.6% vs 33.2%). Poor were hospitalised more in public hospitals; this differential was higher in the more developed (40.7% vs 22.9%) compared with the less developed states (54.3% vs 40.1%) in 2014. In less developed states, the decline in the use of public hospitals was higher for the non-poor than the poor (−25.3% vs −16.7%), while in the more developed states, both non-poor and poor showed a similar decline. The hospitalisation in public hospitals for the older population in the major states of India for 1995–1996, 2004 and 2014 is presented in online supplementary table 4.

**Table 1** Hospitalisation rate per 1000 (95% CI) by age and disease groups in NSS 1995–1996, NSS 2004 and NSS 2014, India

| Age (years) | Hospitalisation rate per 1000 (95% CI) | | | Estimated hospitalised cases (in millions) (%) |
| --- | --- | --- | --- | --- |
| | NCDs | CDs | All diseases | |
| | **NSS 1995–1996** | | | |
| 0–4 | 2.2 (1.8 to 2.6) | 7.8 (7.0 to 8.6) | 14.1 (12.9 to 15.3) | 1.4 (9.7) |
| 5–14 | 2.0 (1.8 to 2.3) | 3.0 (2.7 to 3.3) | 6.8 (6.3 to 7.2) | 1.4 (10.3) |
| 15–29 | 3.6 (3.3 to 3.9) | 6.0 (5.5 to 6.4) | 13.9 (13.2 to 14.7) | 3.1 (22.0) |
| 30–44 | 6.8 (6.3 to 7.3) | 6.0 (5.5 to 6.5) | 17.8 (17.0 to 18.6) | 2.9 (20.5) |
| 45–59 | 14.1 (12.9 to 15.2) | 6.4 (5.7 to 7.2) | 28.0 (26.4 to 29.5) | 2.9 (20.5) |
| 60–69 | 24.4 (22.0 to 26.8) | 8.6 (7.2 to 10.0) | 42.2 (39.2 to 45.2) | 1.2 (8.9) |
| 70 years or more | 35.7 (31.1 to 40.3) | 11.1 (8.5 to 13.7) | 61.8 (55.9 to 67.7) | 1.1 (8.1) |
| Under 60 years | 5.0 (4.8 to 5.2) | 5.5 (5.2 to 5.7) | 14.6 (14.2 to 15.0) | 11.6 (83.0) |
| 60 years or more | 28.7 (26.4 to 31.0) | 9.5 (8.2 to 10.8) | 49.7 (46.8 to 52.7) | 2.4 (17.0) |
| All ages | 6.4 (6.1 to 6.6) | 5.7 (5.5 to 5.9) | 16.6 (16.2 to 17.0) | 14.0 (1.7) |
| | **NSS 2004** | | | |
| 0–4 | 4.4 (3.8 to 4.9) | 15.0 (13.8 to 16.1) | 23.9 (22.5 to 25.4) | 2.6 (9.5) |
| 5–14 | 4.0 (3.6 to 0.5) | 5.6 (5.2 to 6.1) | 11.8 (11.1 to 12.5) | 2.7 (9.9) |
| 15–29 | 10.3 (9.7 to 10.9) | 5.9 (5.5 to 6.4) | 21.4 (20.5 to 22.2) | 5.4 (19.9) |
| 30–44 | 15.8 (15.0 to 16.6) | 7.5 (6.8 to 8.2) | 29.7 (28.5 to 30.9) | 5.7 (21.0) |
| 45–59 | 30.1 (28.6 to 31.6) | 10.5 (9.6 to 11.3) | 47.8 (45.9 to 49.6) | 5.6 (20.5) |
| 60–69 | 45.2 (42.1 to 48.2) | 12.2 (10.7 to 13.8) | 65.7 (62.1 to 69.3) | 2.9 (10.6) |
| 70 years or more | 70.0 (65.0 to 74.9) | 13.7 (11.7 to 15.6) | 95.9 (90.3 to 101.6) | 2.3 (8.5) |
| Under 60 years | 11.7 (11.4 to 12.1) | 7.9 (7.6 to 8.2) | 24.5 (24.0 to 24.9) | 21.9 (80.8) |
| 60 years or more | 54.0 (51.3 to 56.6) | 12.7 (11.5 to 14.0) | 76.4 (73.3 to 79.5) | 5.2 (19.2) |
| All ages | 14.7 (14.4 to 15.1) | 8.3 (8.0 to 8.6) | 28.2 (27.7 to 28.7) | 27.2 (2.8) |
| | **NSS 2014** | | | |
| 0–4 | 8.3 (7.3 to 9.3) | 25.0 (23.3 to 26.7) | 34.2 (32.3 to 36.2) | 3.4 (8.2) |
| 5–14 | 6.6 (5.8 to 7.3) | 7.6 (7.0 to 8.1) | 14.4 (13.5 to 15.4) | 3.3 (7.8) |
| 15–29 | 11.6 (10.8 to 12.4) | 12.2 (11.5 to 12.9) | 24.6 (23.5 to 25.7) | 7.5 (17.9) |
| 30–44 | 22.1 (20.9 to 23.3) | 11.1 (10.2 to 12.1) | 34.6 (33.0 to 36.1) | 8.4 (20.2) |
| 45–59 | 41.7 (39.7 to 43.7) | 13.1 (11.8 to 14.3) | 56.5 (54.2 to 58.9) | 9.2 (22.2) |
| 60–69 | 72.8 (68.0 to 77.7) | 17.1 (15.0 to 19.3) | 92.2 (86.8 to 97.5) | 5.3 (12.7) |
| 70 years or more | 116.2 (107.4 to 124.9) | 20.8 (18.2 to 23.4) | 141.2 (131.9 to 150.5) | 4.6 (11.0) |
| Under 60 years | 17.4 (16.9 to 17.9) | 12.3 (11.9 to 12.7) | 30.7 (30.0 to 31.4) | 31.8 (76.4) |
| 60 years or more | 88.5 (84.1 to 92.9) | 18.4 (16.8 to 20.1) | 109.9 (105.1 to 114.7) | 9.8 (23.6) |
| All ages | 23.1 (22.5 to 23.7) | 12.8 (12.4 to 13.2) | 37.0 (36.3 to 37.7) | 41.6 (3.7) |

CD, Communicable diseases and nutritional disorder; NCD, Non-communicable diseases and injuries; NSS, National Sample Survey.

All subgroups of the older population showed a significant increase in hospitalisation rates, but there was considerable variation in the amount of change (table 4). Between 1995 and 2014, the increase in hospitalisation rate was higher for females (2.82 vs 1.87 times), single (3.04 vs 1.89 times), poor (2.72 vs 1.87 times), illiterate (2.45 vs 1.77 times), rural residents (2.32 vs 1.88 times) and those living in the less developed states (3.07 vs 1.95 times) compared with their respective counterparts. This reduced the differential in hospitalisation rate by gender, marital status, economic status, place of residence and states.

### Compositional change
Most of the older population lived in rural areas, but their proportion decreased by 9.3 percentage points (78.1%–68.8%) between 1995 and 2014 (table 5). There was 5.2 percentage points (58.3% in 1995–1996 to 63.4% in 2014) increase in the proportion of currently married older population. Literacy in the

older population increased by 13.0 percentage points by 2014. In 1995–1996, most of the older population were physically mobile (89.5%), <70 years of age (62.5%), resident of the more developed states (53.7%), economically dependent (68.9%) and reported good SRH (80.8%), with only marginal change in their proportions. The majority of the older population were non-scheduled castes (SC)/scheduled tribes (STs) (76.4%), poor (64.2%), living with family (95.6%) and reporting better or nearly same SRH compared with past year (74.3%) in 1995–1996 and their proportion remained unchanged in 2014.

### Determinants of hospitalisation

Older population reporting poor SRH (adjusted odds ratio (AOR) 2.42 95% CI 1.91 to 3.07) and living alone (AOR 2.13 95% CI 1.44 to 3.16) had the highest odds of hospitalisation in 1995–1996 and 2014, respectively (table 6). Poor older population were 59% (95% CI 0.35 to 0.48) and 37% (95% CI 0.55 to 0.72) less likely to be hospitalised in 1995–1996 and 2014, respectively. The economically dependent older population was 32% (95% CI 1.08 to 1.62) more likely to be hospitalised in

1995–1996. Older population living in the less developed states had lower odds of hospitalisation in 1995–1996 (AOR 0.34 95% CI 0.29 to 0.40) and 2014 (AOR 0.54 95% CI 0.47 to 0.61). In 1995–1996, female and single older population were 30% (95% CI 0.60 to 0.83) and 34% (95% CI 0.57 to 0.77) less likely to be hospitalised, respectively. The older population belonging to SC/STs had lower odds of hospitalisation (AOR 0.81, 95% CI 0.70 to 0.94) compared with non-SC/STs in 2014. In 2014, physically immobile and those reporting SRH worse than previous year had 85% (95% CI 1.15 to 2.27) and 67% (95% CI 1.44 to 1.94) higher odds of being hospitalised, respectively. After adjusting for the covariates, age and place of residence were not significantly associated with hospitalisation.

Between 1995 and 2014, there was a modest increase in intercept for the outcome variable suggesting that when all the explanatory variables in the logit model were set equal to their reference categories, the probability of hospitalisation was significantly higher in 2014 than in 1995–1996 for the older population. Comparison of 1995–1996 and 2014 coefficients showed the convergence

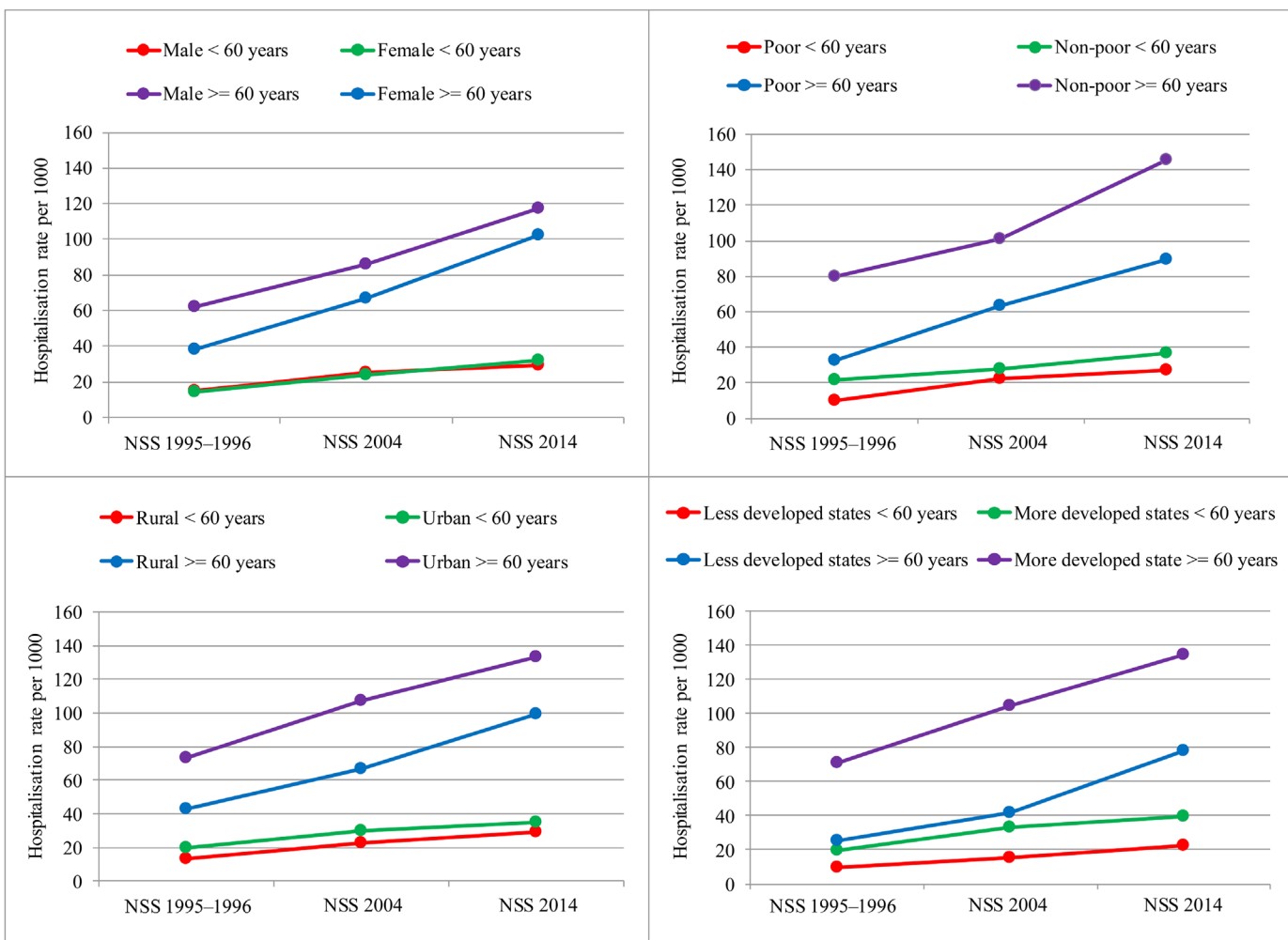

**Figure 1** Socioeconomic and demographic differentials in hospitalisation rates in NSS 1995–1996, NSS 2004 and NSS 2014, India.

**Table 2** Hospitalisation rate per 1000 (95% CI) by disease groups in the less developed and more developed states in NSS 1995–1996, NSS 2004 and NSS 2014, India

Hospitalisation rate per 1000 (95% CI)

### 60 years or more

| States | NSS 1995–1996 | | | NSS 2004 | | | NSS 2014 | | |
| --- | --- | --- | --- | --- | --- | --- | --- | --- | --- |
| | All hospitalisations | NCDs | CDs | All hospitalisations | NCDs | CDs | All hospitalisations | NCDs | CDs |
| Less developed | 25.1 (22.3 to 27.9) | 13.6 (12.1 to 15.1) | 5.8 (4.0 to 7.6) | 41.6 (38.4 to 44.9) | 28.6 (25.8 to 31.4) | 7.3 (6.2 to 8.4) | 78.4 (71.3 to 85.5) | 61.2 (54.6 to 67.8) | 15.0 (12.7 to 17.2) |
| More developed | 70.9 (66.1 to 75.8) | 41.7 (37.7 to 45.8) | 12.7 (10.8 to 14.6) | 104.6 (99.8 to 109.4) | 74.6 (70.4 to 78.7) | 17.1 (15.1 to 19.1) | 134.3 (128.0 to 140.7) | 109.7 (103.9 to 115.5) | 21.1 (18.8 to 23.5) |
| India | 49.7 (46.8 to 52.6) | 28.7 (26.5 to 31.0) | 9.5 (8.2 to 10.8) | 76.4 (73.4 to 79.4) | 54.0 (51.4 to 56.5) | 12.7 (11.5 to 13.9) | 109.9 (105.2 to 114.5) | 88.5 (84.2 to 92.8) | 18.4 (16.8 to 20.1) |

### Under 60 years

| States | NSS 1995–1996 | | | NSS 2004 | | | NSS 2014 | | |
| --- | --- | --- | --- | --- | --- | --- | --- | --- | --- |
| | All hospitalisations | NCDs | CDs | All hospitalisations | NCDs | CDs | All hospitalisations | NCDs | CDs |
| Less developed | 9.4 (8.9 to 9.8) | 2.9 (2.7 to 3.1) | 3.7 (3.4 to 4.0) | 15.7 (15.2 to 16.1) | 7.3 (7.0 to 7.6) | 5.2 (4.9 to 5.4) | 22.3 (21.5 to 23.1) | 11.8 (11.2 to 12.4) | 9.9 (9.4 to 10.4) |
| More developed | 19.5 (18.9 to 20.1) | 7.0 (6.6 to 7.3) | 7.1 (6.7 to 7.4) | 33.1 (32.3 to 34.0) | 16.1 (15.5 to 16.7) | 10.5 (10.0 to 11.1) | 39.9 (38.8 to 40.9) | 23.5 (22.6 to 24.4) | 15.0 (14.3 to 15.6) |
| India | 14.6 (14.2 to 15.0) | 5.0 (4.8 to 5.2) | 5.5 (5.2 to 5.7) | 24.5 (24.0 to 24.9) | 11.7 (11.4 to 12.1) | 7.9 (7.6 to 8.2) | 30.7 (30.0 to 31.4) | 17.4 (16.9 to 17.9) | 12.3 (11.9 to 12.7) |

### All ages

| States | NSS 1995–1996 | | | NSS 2004 | | | NSS 2014 | | |
| --- | --- | --- | --- | --- | --- | --- | --- | --- | --- |
| | All hospitalisations | NCDs | CDs | All hospitalisations | NCDs | CDs | All hospitalisations | NCDs | CDs |
| Less developed | 10.2 (9.8 to 10.6) | 3.5 (3.3 to 3.7) | 3.8 (3.6 to 4.1) | 17.5 (17.0 to 18.0) | 8.7 (8.4 to 9.0) | 5.4 (5.1 to 5.6) | 26.1 (25.2 to 27.0) | 15.2 (14.4 to 15.9) | 10.2 (9.7 to 10.7) |
| More developed | 22.5 (21.9 to 23.1) | 9.0 (8.6 to 9.4) | 7.4 (7.0 to 7.7) | 38.7 (37.8 to 39.6) | 20.6 (20.0 to 21.3) | 11.1 (10.6 to 11.6) | 48.6 (47.5 to 49.8) | 31.5 (30.5 to 32.4) | 15.6 (14.9 to 16.2) |
| India | 16.6 (16.2 to 17.0) | 6.4 (6.1 to 6.6) | 5.7 (5.5 to 5.9) | 28.2 (27.7 to 28.7) | 14.7 (14.4 to 15.1) | 8.3 (8.0 to 8.6) | 37.0 (36.3 to 37.7) | 23.1 (22.5 to 23.7) | 12.8 (12.4 to 13.2) |

CD, Communicable diseases and nutritional disorder; NCD, Non-communicable diseases and injuries; NSS, National Sample Survey.

**Table 3** Hospitalisation rate per 1000 (95% CI) in public hospitals by economic status in the less developed and more developed states in NSS 1995–1996, NSS 2004 and NSS 2014, India

Hospitalisation rate per 1000 (95% CI) in public hospitals

**60years or more**

| States | NSS 1995–1996 | | | NSS 2004 | | | NSS 2014 | | |
|---|---|---|---|---|---|---|---|---|---|
| | Non-poor | Poor | Total | Non-poor | Poor | Total | Non-poor | Poor | Total |
| Less developed | 53.3 (45.6 to 60.8) | 64.8 (56.0 to 72.7) | 57.1 (51.3 to 62.6) | 38.7 (33.6 to 44.2) | 59.5 (54.9 to 63.9) | 48.9 (45.0 to 52.9) | 36.0 (30.4 to 41.9) | 55.0 (48.9 to 60.9) | 45.2 (40.9 to 49.6) |
| More developed | 27.2 (23.6 to 31.1) | 52.4 (46.9 to 57.8) | 38.5 (35.0 to 42.1) | 28.1 (25.0 to 31.3) | 42.6 (39.4 to 45.8) | 36.1 (33.9 to 38.4) | 20.7 (18.0 to 23.6) | 41.1 (38.2 to 44.1) | 31.6 (29.5 to 33.8) |
| India | 34.1 (30.4 to 37.9) | 54.6 (49.9 to 59.2) | 42.7 (39.7 to 45.8) | 30.9 (28.3 to 33.6) | 46.3 (43.6 to 49.1) | 39.2 (37.3 to 41.2) | 25.8 (23.2 to 28.4) | 45.2 (42.5 to 47.9) | 35.9 (33.9 to 37.8) |

**Under 60years**

| States | NSS 1995–1996 | | | NSS 2004 | | | NSS 2014 | | |
|---|---|---|---|---|---|---|---|---|---|
| | Non-poor | Poor | Total | Non-poor | Poor | Total | Non-poor | Poor | Total |
| Less developed | 53.8 (51.1 to 56.4) | 65.3 (60.6 to 69.7) | 58.0 (55.6 to 60.4) | 43.5 (41.4 to 45.6) | 51.7 (49.6 to 53.8) | 47.8 (46.3 to 49.3) | 41.3 (38.7 to 43.9) | 54.2 (51.7 to 56.7) | 48.2 (46.4 to 50.0) |
| More developed | 30.0 (28.3 to 31.9) | 51.9 (49.6 to 54.2) | 40.0 (38.5 to 41.5) | 28.1 (26.4 to 29.9) | 44.1 (42.4 to 45.8) | 38.0 (36.7 to 39.2) | 23.7 (21.8 to 25.6) | 40.6 (38.9 to 42.3) | 33.7 (32.4 to 35.1) |
| India | 37.9 (36.3 to 39.4) | 55.3 (53.2 to 57.4) | 45.4 (44.1 to 46.7) | 33.8 (32.4 to 35.1) | 46.2 (44.9 to 47.6) | 41.1 (40.1 to 42.1) | 30.9 (29.4 to 32.5) | 45.4 (44.0 to 46.9) | 39.2 (38.2 to 40.3) |

**All ages**

| States | NSS 1995–1996 | | | NSS 2004 | | | NSS 2014 | | |
|---|---|---|---|---|---|---|---|---|---|
| | Non-poor | Poor | Total | Non-poor | Poor | Total | Non-poor | Poor | Total |
| Less developed | 53.7 (51.2 to 56.2) | 65.2 (61.0 to 69.2) | 57.9 (55.7 to 60.0) | 42.5 (40.5 to 44.5) | 52.5 (50.6 to 54.5) | 47.7 (46.3 to 49.1) | 40.1 (37.7 to 42.6) | 54.3 (52.0 to 56.6) | 47.6 (45.9 to 49.3) |
| More developed | 29.5 (27.9 to 31.1) | 52.0 (49.8 to 54.1) | 39.7 (38.3 to 41.1) | 28.0 (26.5 to 29.6) | 43.7 (42.3 to 45.3) | 37.5 (36.4 to 38.6) | 22.9 (21.3 to 24.5) | 40.7 (57.8 to 60.7) | 33.2 (32.1 to 34.3) |
| India | 37.2 (35.8 to 38.7) | 55.2 (53.3 to 57.1) | 44.9 (43.7 to 46.1) | 33.1 (31.9 to 34.3) | 46.2 (44.9 to 47.4) | 40.6 (39.8 to 41.5) | 29.6 (28.3 to 31.0) | 45.4 (44.1 to 46.6) | 38.4 (37.5 to 39.4) |

NSS, National Sample Survey.

**Table 4**  Hospitalisation rate per 1000 (95% CI) for older population by background characteristics in NSS 1995–1996, NSS 2004 and NSS 2014, India

| Background characteristics | Hospitalisation rate per 1000 (95% CI) | | |
|---|---|---|---|
| Predisposing variables | NSS 1995–1996 | NSS 2004 | NSS 2014 |
| Age (years) | | | |
| 60–69 | 37.6 (34.8 to 40.5) | 62.2 (58.8 to 65.6) | 82.6 (77.6 to 87.6) |
| 70 years or more | 53.1 (47.8 to 58.4) | 90.6 (85.3 to 96.0) | 124.4 (116.4 to 132.4) |
| Sex | | | |
| Male | 53.9 (49.3 to 58.4) | 80.3 (76.3 to 84.2) | 101.0 (95.5 to 106.6) |
| Female | 33.3 (30.4 to 36.1) | 63.7 (59.5 to 67.9) | 94.0 (87.5 to 100.5) |
| Marital status | | | |
| Currently married | 50.8 (46.8 to 54.9) | 75.6 (72.0 to 79.1) | 95.9 (91.2 to 100.7) |
| Single | 32.9 (29.8 to 36.0) | 66.8 (61.9 to 71.6) | 100.1 (91.8 to 108.4) |
| Caste | | | |
| Non-SC/STs | 46.7 (43.5 to 50.0) | 78.8 (75.3 to 82.2) | 105.2 (100.0 to 110.4) |
| SC/STs | 32.9 (28.4 to 37.3) | 50.7 (45.8 to 55.5) | 71.8 (65.8 to 77.9) |
| Education | | | |
| Literate | 65.9 (60.7 to 71.1) | 106.3 (100.6 to 112.0) | 116.7 (110.2 to 123.2) |
| Illiterate | 34.0 (30.9 to 37.2) | 54.2 (50.9 to 57.5) | 83.2 (77.5 to 88.8) |
| **Enabling variables** | | | |
| Place of residence | | | |
| Urban | 63.1 (58.7 to 67.4) | 99.5 (92.8 to 106.3) | 118.6 (111.2 to 126.0) |
| Rural | 37.9 (34.7 to 41.1) | 63.2 (60.0 to 66.3) | 87.8 (82.6 to 93.1) |
| States | | | |
| More developed | 62.1 (57.8 to 66.5) | 98.4 (93.8 to 103.0) | 121.0 (114.9 to 127.1) |
| Less developed | 21.8 (19.0 to 24.5) | 39.5 (36.4 to 42.6) | 67.0 (61.2 to 72.9) |
| Economic dependency | | | |
| Economically independent | 35.8 (30.9 to 40.8) | 63.2 (58.9 to 67.5) | 89.2 (80.2 to 98.2) |
| Economically dependent | 47.2 (44.0 to 50.4) | 77.9 (74.1 to 81.7) | 100.7 (96.0 to 105.5) |
| Economic status | | | |
| Non-poor | 68.6 (62.6 to 74.6) | 94.9 (89.2 to 100.6) | 128.2 (119.1 to 137.4) |
| Poor | 29.4 (26.9 to 31.9) | 59.8 (56.5 to 63.0) | 80.1 (75.8 to 84.3) |
| Living arrangement | | | |
| With family | 44.2 (41.4 to 47.0) | 74.1 (71.1 to 77.1) | 95.3 (91.4 to 99.3) |
| Alone | 31.1 (22.2 to 40.0) | 54.0 (41.1 to 67.0) | 146.2 (99.3 to 193.2) |
| **Need variables** | | | |
| Physical mobility status | | | |
| Mobile | 38.0 (35.4 to 40.7) | 62.5 (59.8 to 65.3) | 84.3 (80.3 to 88.3) |
| Immobile | 91.3 (78.8 to 103.7) | 193.9 (175.0 to 212.8) | 249.4 (222.3 to 276.5) |
| Current self-rated health (SRH) | | | |
| Good | 31.2 (28.9 to 33.4) | 54.3 (51.5 to 57.1) | 67.8 (63.8 to 71.7) |
| Poor | 96.9 (86.4 to 107.4) | 138.3 (129.5 to 147.1) | 200.2 (186.8 to 213.7) |
| SRH compared with previous year | | | |
| Better or same | 31.9 (29.4 to 34.5) | 57.4 (54.6 to 60.1) | 70.1 (66.0 to 74.3) |
| Worse | 78.3 (70.7 to 85.9) | 138.9 (128.9 to 148.9) | 179.5 (167.8 to 191.2) |
| Total | 43.4 (40.8 to 46.1) | 72.0 (69.1 to 74.8) | 97.5 (93.2 to 101.7) |

NSS, National Sample Survey; SC/STs, Scheduled castes/scheduled tribes.

**Table 5** Background characteristics of the older population in NSS 1995–1996, NSS 2004 and NSS 2014, India

| Background characteristics | NSS 1995–1996 | | NSS 2004 | | NSS 2014 | |
|---|---|---|---|---|---|---|
| **Predisposing variables** | **N** | **% (95% CI)** | **N** | **% (95% CI)** | **N** | **% (95% CI)** |
| Age (years) | | | | | | |
| 60–69 | 21 124 | 62.5 (61.6 to 63.4) | 22 546 | 65.3 (64.6 to 66.0) | 17 160 | 64.5 (63.2 to 65.8) |
| 70 years or more | 12 866 | 37.5 (36.6 to 38.4) | 12 264 | 34.7 (34.0 to 35.4) | 10 085 | 35.5 (34.2 to 36.8) |
| Sex | | | | | | |
| Male | 17 173 | 49.4 (48.5 to 50.4) | 17 750 | 50.0 (49.3 to 50.8) | 13 692 | 49.2 (47.8 to 50.6) |
| Female | 16 817 | 50.6 (49.6 to 51.5) | 17 081 | 50.0 (49.2 to 50.7) | 13 553 | 50.8 (49.4 to 52.2) |
| Marital status | | | | | | |
| Currently married | 20 111 | 58.3 (57.3 to 59.2) | 20 959 | 59.2 (58.5 to 60.0) | 17 947 | 63.4 (62.1 to 64.7) |
| Single | 13 852 | 41.7 (40.8 to 42.7) | 13 872 | 40.8 (40.0 to 41.5) | 9298 | 36.6 (35.3 to 37.9) |
| Caste | | | | | | |
| Non-SC/STs | 26 089 | 76.4 (75.6 to 77.2) | 26 291 | 76.0 (75.3 to 76.6) | 20 823 | 76.8 (75.6 to 77.9) |
| SC/STs | 7880 | 23.6 (22.8 to 24.4) | 8531 | 24.0 (23.4 to 24.7) | 6422 | 23.2 (22.1 to 24.4) |
| Education | | | | | | |
| Literate | 12 406 | 29.5 (28.7 to 30.4) | 13 514 | 34.2 (33.5 to 34.9) | 13 362 | 42.6 (41.2 to 43.9) |
| Illiterate | 21 543 | 70.5 (69.6 to 71.3) | 21 301 | 65.8 (65.1 to 66.5) | 13 883 | 57.4 (56.1 to 58.8) |
| **Enabling variables** | | | | | | |
| Place of residence | | | | | | |
| Urban | 13 035 | 21.9 (21.3 to 22.5) | 12 566 | 24.3 (23.7 to 24.9) | 12 226 | 31.2 (30.0 to 32.4) |
| Rural | 20 955 | 78.1 (77.5 to 78.7) | 22 265 | 75.7 (75.1 to 76.3) | 15 019 | 68.8 (67.6 to 70.0) |
| States | | | | | | |
| More developed | 17 389 | 53.7 (52.8 to 54.7) | 17 019 | 55.2 (54.4 to 55.9) | 14 466 | 56.3 (54.9 to 57.6) |
| Less developed | 16 601 | 46.3 (45.3 to 47.2) | 17 812 | 44.8 (44.1 to 45.6) | 12 779 | 43.7 (42.4 to 45.1) |
| Economic dependency | | | | | | |
| Economically independent | 10 149 | 31.1 (30.2 to 32.0) | 11 800 | 34.0 (33.3 to 34.7) | 7159 | 28.3 (27.0 to 29.6) |
| Economically dependent | 23 061 | 68.9 (68.0 to 69.8) | 22 429 | 66.0 (65.3 to 66.7) | 20 075 | 71.7 (70.4 to 73.0) |
| Economic status | | | | | | |
| Non-poor | 15 407 | 35.8 (35.0 to 36.7) | 14 372 | 34.8 (34.1 to 35.5) | 11 738 | 36.1 (34.8 to 37.4) |
| Poor | 18 583 | 64.2 (63.3 to 65.0) | 20 459 | 65.2 (64.5 to 65.9) | 15 507 | 63.9 (62.6 to 65.2) |
| Living arrangement | | | | | | |
| With family | 32 482 | 95.6 (95.2 to 96.0) | 32 595 | 94.8 (94.4 to 95.1) | 26 659 | 95.9 (95.3 to 96.5) |
| Alone | 1174 | 4.4 (4.0 to 4.8) | 1509 | 5.2 (4.9 to 5.6) | 586 | 4.1 (3.5 to 4.7) |
| **Need variables** | | | | | | |
| Physical mobility status | | | | | | |
| Mobile | 29 697 | 89.5 (88.9 to 90.1) | 30 821 | 91.9 (91.5 to 92.3) | 24 499 | 92.0 (91.3 to 92.7) |
| Immobile | 3635 | 10.5 (9.9 to 11.1) | 3224 | 8.1 (7.7 to 8.5) | 2735 | 8.0 (7.3 to 8.7) |
| Current self-rated health (SRH) | | | | | | |
| Good | 27 263 | 80.8 (79.9 to 81.5) | 24 965 | 76.4 (75.7 to 77.0) | 20 143 | 77.6 (76.4 to 78.7) |
| Poor | 6217 | 19.2 (18.5 to 20.1) | 8216 | 23.6 (23.0 to 24.3) | 7091 | 22.4 (21.3 to 23.6) |
| SRH compared with previous year | | | | | | |
| Better or same | 25 018 | 74.3 (73.4 to 75.1) | 25 971 | 79.3 (78.7 to 79.9) | 19 590 | 75.0 (73.8 to 76.2) |
| Worse | 8430 | 25.7 (24.9 to 26.6) | 7210 | 20.7 (20.1 to 21.3) | 7644 | 25.0 (23.8 to 26.2) |
| N | 33 990 | | 34 831 | | 27 245 | |

NSS, National Sample Survey; SC/STs, Scheduled castes/scheduled tribes.

**Table 6** Determinants of hospitalisation for the older population in NSS 1995–1996 and NSS 2014, India

| Background characteristics | Whether hospitalised | | | | | | | |
|---|---|---|---|---|---|---|---|---|
| | $\beta_{1995–1996}$ | Exp ($\beta_{1995–1996}$) | 95% CI for Exp ($\beta_{1995–1996}$) | $\beta_{2014}$ | Exp ($\beta_{2014}$) | 95% CI for Exp ($\beta_{2014}$) | $\beta_{2014} - \beta_{1995–1996}$ | P value for Wald test ($\beta_{2014} - \beta_{1995–1996}$) |
| Predisposing variables | | | | | | | | |
| Age (years) (ref.=60–69) | | | | | | | | |
| 70 years or more | −0.028 | 0.97 | (0.83 to 1.14) | 0.124 | 1.13 | (0.99 to 1.29) | 0.152 | 0.147 |
| Sex (ref.=male) | | | | | | | | |
| Female | −0.352 | 0.70 | (0.60 to 0.83) | −0.050 | 0.95 | (0.83 to 1.10) | 0.302 | 0.006 |
| Marital status (ref.=currently married) | | | | | | | | |
| Single | −0.416 | 0.66 | (0.57 to 0.77) | −0.130 | 0.88 | (0.76 to 1.02) | 0.286 | 0.009 |
| Caste (ref.=non-SC/STs) | | | | | | | | |
| SC/STs | 0.017 | 1.02 | (0.84 to 1.23) | −0.211 | 0.81 | (0.70 to 0.94) | −0.229 | 0.060 |
| Literacy status (ref.=literate) | | | | | | | | |
| Illiterate | −0.278 | 0.76 | (0.63 to 0.91) | −0.224 | 0.80 | (0.70 to 0.92) | 0.055 | 0.645 |
| Enabling variables | | | | | | | | |
| Place of residence (ref.=urban) | | | | | | | | |
| Rural | −0.112 | 0.89 | (0.76 to 1.04) | −0.032 | 0.97 | (0.85 to 1.11) | 0.080 | 0.446 |
| States (ref.= more developed) | | | | | | | | |
| Less developed | −1.070 | 0.34 | (0.29 to 0.40) | −0.619 | 0.54 | (0.47 to 0.61) | 0.451 | <0.001 |
| Economic dependence (ref.= independent) | | | | | | | | |
| Economically dependent | 0.281 | 1.32 | (1.08 to 1.62) | 0.004 | 1.00 | (0.85 to 1.18) | −0.277 | 0.035 |
| Economic status (ref.=non-poor) | | | | | | | | |
| Poor | −0.895 | 0.41 | (0.35 to 0.48) | −0.462 | 0.63 | (0.55 to 0.72) | 0.432 | <0.001 |
| Living arrangement (ref.=living with family) | | | | | | | | |
| Living alone | 0.197 | 1.22 | (0.85 to 1.74) | 0.757 | 2.13 | (1.44 to 3.16) | 0.560 | 0.039 |
| Need variables | | | | | | | | |
| Physical mobility status (ref.= mobile) | | | | | | | | |
| Immobile | 0.400 | 1.49 | (1.21 to 1.84) | 0.617 | 1.85 | (1.51 to 2.27) | 0.217 | 0.149 |
| Current self-rated health (ref.=good SRH) | | | | | | | | |
| Poor SRH | 0.884 | 2.42 | (1.91 to 3.07) | 0.736 | 2.09 | (1.78 to 2.44) | −0.149 | 0.306 |
| SRH compared with last year (ref.=better or nearly the same) | | | | | | | | |
| Worse SRH | 0.475 | 1.61 | (1.31 to 1.98) | 0.515 | 1.67 | (1.44 to 1.94) | 0.039 | 0.763 |
| Constant | −2.466 | 0.08 | (0.07 to 0.10) | −2.238 | 0.11 | (0.09 to 0.12) | 0.228 | 0.037 |

Continued

**Table 6** Continued

| Background characteristics | Whether hospitalised | | | | | | |
| --- | --- | --- | --- | --- | --- | --- | --- |
| | $\beta_{1995-1996}$ | Exp ($\beta_{1995-1996}$) | 95% CI for Exp ($\beta_{1995-1996}$) | $\beta_{2014}$ | Exp ($\beta_{2014}$) | 95% CI for Exp ($\beta_{2014}$) | P value for Wald test ($\beta_{2014}-\beta_{1995-1996}$) |
| F-adjusted test statistic | 1.61 | | | 0.81 | | | |
| P value | 0.106 | | | 0.611 | | | |
| N | 32 780 | | | 27 234 | | | |

NSS, National Sample Survey; SC/STs, Scheduled castes/scheduled tribes.

of differentials in hospitalisation by gender, marital status, economic status, living arrangement and states (table 6).

### Decomposition of increase in hospitalisation rate

For the older population in India, the propensity change explained 86.6% of the increase in hospitalisation rate between 1995 and 2014 (table 7). The improved propensity to use hospital care by economically poor, residents of the less developed states, females and singles contributed 16.4%, 12.3%, 9.0% and 7.1% of the increase in hospitalisation rate, respectively, regardless of the change in their composition. The change in intercept accounted for 13.5% of the increase in hospitalisation rate. Change in the composition of the characteristics of older population had a modest influence on the level of hospitalisation; contributing 9.2% of the increase in hospitalisation. Many of the changes in the population structure during the intersurvey period favoured increased hospitalisation, except gender and physical mobility status. The increase in the proportion of literates, those reporting poor SRH, economically dependent and single contributed 2.1%, 1.7%, 1.6% and 1.3% of the increase in hospitalisation rate, respectively between 1995 and 2014, regardless of the change in the likelihood of hospitalisation by the subgroups.

### DISCUSSION

This report provides evidence on trends in hospitalisation rates in India over two decades up to 2014, and compares the older population with population under 60 years. Five key findings relating to hospitalisation trends and differentials emerge from this study. First, the hospitalisation rate increased twofold between 1995 and 2014; the increase was higher for NCDs and in less developed states. Second, poor people used more public hospitals; this differential was higher in the more developed than the less developed states. Third, the older population had higher hospitalisation rates and greater proportion of hospitalisation for NCDs than the population under 60 years. Fourth, among the older population, the hospitalisation rate was comparatively lower for females, poor, and rural residents. Fifth, propensity change was largely responsible for the increase in hospitalisation among the older population in India over these two decades.

Hospitalisation is an important indicator of the demand for curative care and is an integral part of any health system. The increase in hospitalisation rate found in our study could be due to the growing awareness about the health prevention and other precautionary measures along with proper diagnosis of the health conditions. The evidence on increasing hospitalisation is vital for planning of resources to meet the growing demand for inpatient care and for formulating viable publicly funded financial risk protection mechanism. To provide targeted financial protective intervention, it would also be useful to know whether the increase in hospitalisation was due to higher hospitalisations for preventive care among the

**Table 7** Decomposition of increase in hospitalisation for the older population between NSS 1995–1996 and NSS 2014, India

| Background characteristics | Contribution to the increase in hospitalisation (%)* | | |
| --- | --- | --- | --- |
| | **Propensity** | **Composition** | **Interaction** |
| 70 years or more | 0.06 (3.4) | 0.00 (0.0) | 0.00 (−0.2) |
| Female | 0.15 (9.0) | 0.00 (−0.1) | 0.00 (0.0) |
| Single | 0.12 (7.1) | 0.02 (1.3) | −0.01 (−0.9) |
| SC/STs | −0.05 (−3.2) | 0.00 (0.0) | 0.00 (0.0) |
| Illiterate | 0.04 (2.3) | 0.04 (2.1) | −0.01 (−0.4) |
| Rural | 0.06 (3.7) | 0.01 (0.6) | −0.01 (−0.4) |
| Less developed states | 0.21 (12.3) | 0.03 (1.6) | −0.01 (−0.7) |
| Economically dependent | −0.19 (−11.3) | 0.01 (0.5) | −0.01 (−0.5) |
| Economically poor | 0.28 (16.4) | 0.00 (0.1) | 0.00 (−0.1) |
| Living alone | 0.02 (1.4) | 0.00 (0.0) | 0.00 (−0.1) |
| Physically immobile | 0.02 (1.3) | −0.01 (−0.6) | −0.01 (−0.3) |
| Poor SRH | −0.03 (−1.7) | 0.03 (1.7) | 0.00 (−0.3) |
| Worse SRH than previous year | 0.01 (0.6) | 0.00 (−0.2) | 0.00 (0.0) |
| Intercept | 0.23 (13.5) | | |
| % contribution to the overall increase | 86.6 | 9.2 | 4.2 |

*Per cent contribution has been calculated as the ratio of the contribution of the covariate and the sum of the absolute contribution of covariates under the propensity, composition and interaction components multiplied by 100.

NSS, National Sample Survey; SC/STs, Scheduled castes/scheduled tribes.

rich or emergency inpatient care among the poor. Data from the global burden of disease study suggests that of the total disease burden, measured as disability-adjusted life years lost in India, the contribution of non-communicable disease and injuries has increased from 38.4% in 1990 to 64.2% in 2013.[33] The higher increase in hospitalisation for NCDs over two decades is consistent with the shifting disease burden trends in India.

The developed states in India with good health indicators are usually found to report higher use of healthcare.[10 22] Higher hospitalisation rate in the more developed states of India may indicate a higher volume of health services provided by health sector, rather than reflect higher morbidity prevalence. Interestingly, we found that the increase in hospitalisation rate between 1995 and 2014 was more pronounced in the less developed than the more developed states. A plausible reason for this could be the increased burden of chronic, degenerative and lifestyle diseases in the less developed states because of their advancement through the health transition process. Other factors contributing to this could be the greater availability of health services, better access to healthcare or the increased propensity to use healthcare.

The increase in the use of private hospitals over two decades in India is a matter of concern from the equity point of view and has cost implications for the poor. The continuing inadequacies of the public health system and the unrestricted growth of private providers are possible reasons for the decline in the use of public hospitals. The decline in the use of public hospitals was found to be higher for the non-poor in the less developed states,

which implies that in spite of decline, the poor in the less developed states still largely use public hospitals. The increasing provision of inpatient care in private hospitals and the consequent decline in the utilisation of public hospitals is likely to impose a higher financial risk on individuals and households.[34 35] Strengthening the public funding model of service delivery in India would increase the ability of public facilities to meet the increasing demand for healthcare and thereby improve the utilisation of inpatient care by the poor.

Our results indicated clear distinction in levels and differentials in hospitalisation rate between older population and population under 60 years. The older population had more than three times higher hospitalisation than any other age groups. Contributing 8.6% to India's population, older population accounted for nearly one-quarter of all hospital stays in 2014. The improved longevity coupled with the increased years of poor health at older ages is predominantly responsible for the difference between the hospitalisation rates of the two age groups. Data from the global burden of disease study suggest that in India in 1990, disease burden among the older population accounted for 11.8% of the total disease burden. In 2013, this burden had increased to 22.3% of the total disease burden, and non-communicable diseases and injuries made up 82.3% of the total disease burden.[33] Our results showed that the contribution of the older population in total hospitalisation increased over two decades, and they had higher hospitalisation rates for NCDs in any given year. However, the hospitalisations in absolute number and their contribution in total

hospitalisations remain higher for the population under 60 years. Evidence suggests that over the past 25 years the burden of premature death and health loss from NCDs such as heart disease, stroke, chronic obstructive pulmonary disease and road traffic injuries has increased substantially, while the burden due to lower respiratory infections, tuberculosis, diarrhoea and neonatal disorders remains high in India.[33] For the purpose of planning of the resources for universal health coverage and reducing premature mortality, it is important to continue focusing on the child and adult population which account for majority of India's population. At the same time, given the increasing proportion of older population it is equally important to allocate resources and provide healthcare services to cater to their specific healthcare needs.

In the population under 60 years, there was no evidence for gender differential, while, in the older population, a higher proportion of males were hospitalised. Studies from the high-income nations have also found that the older women have less hospital stays than their male counterparts.[15 36–39] Greater economic dependency among females at older ages is a major driver of the gender differential in healthcare use in India.[20] On a positive note, we found that the improved likelihood of using hospital care by female older population contributed to the decline in gender differential among the older population.

In the absence of a health financing system, low level of health insurance coverage and high out-of-pocket cost of healthcare, economic status becomes an important factor affecting healthcare use. We found that the non-poor had higher hospitalisation rates than the poor; this differential was higher for the older population than the other ages. Based on the Andersen's model of healthcare use, we found that the poor older population had significantly less likelihood of using hospital care even after controlling for health profiles. The economic inequality in hospitalisation among the older population is evident in India.[16] Older population rely more on family and other social structures for financial support, and therefore, they might not have adequate resources for hospital care. Financial empowerment of the poor older population can be one way of effectively improving the healthcare utilisation.

An important finding of this study is that the propensity change has contributed most to the twofold increase in hospitalisation of the older population in India between 1995 and 2014. A plausible explanation could be better awareness of the medical conditions and health among the population.[40] A relatively higher increase in hospitalisation among the poor compared with the non-poor older population has contributed most to the increase in hospitalisation rate attributed to propensity change. This indicates a decline in the differentials in healthcare use by economic status over two decades. It has been argued that lowering of inequality will not make the situation more equitable for the poor, if there is a high increase in the rate of hospitalisation, a decline in dependence on government hospitals and a steep hike in the cost of hospital care.[22]

The increase in hospitalisation rate was moderately influenced by the factors not explicitly considered in the model. The supply side factors like the expansion of private healthcare market and consequent improvement in the availability of health services could have propelled the use of healthcare.[22] The expansion of morbidity, with a heavier and cumulated concentration of chronic diseases at older ages, could be another potential driver of the increase in hospitalisation.[41 42] Compositional change contributed marginally to the increase in hospitalisation of the older population over the two decades. It would be interesting to see how the anticipated compositional change influences the future demand for hospitalisation.

The findings of this report must be interpreted in the light of some limitations. First, we used individual determinants and did not examine the full array of determinants of healthcare use as suggested by the Andersen's model of healthcare use. Data on the supply side of healthcare provision were not available from the national sample surveys, nor were comparable data available from other secondary sources corresponding to the survey time points. Second, the use of self-reported data on diseases from the national sample surveys may be associated with biases. However, we report hospitalisation trends for broad groups of diseases which may be reasonable. Even with these limitations, this study uses large-scale data from the nationwide surveys in India over two decades to provide insights into the changing hospitalisation rate by age groups, and the reasons behind the increased hospitalisation of the older population. Given the anticipated further increase of the older population and their higher demand for healthcare, it is time for the policy makers to pay particular attention to planning how adequate resources and mechanisms can be put in place for the provision of geriatric healthcare in India.

**Correction notice** This article has been corrected since it published Online First. Public Health Foundation of India has been added to Lalit Dandona's affiliations.

**Contributors** AP extracted the data, conducted statistical analysis, interpreted the findings and wrote the first draft of the manuscript. GBP contributed to the initial concept of the paper and guided the statistical analysis. LC provided critical comments on the manuscript for intellectual content. LD provided detailed guidance on the study design, analysis, interpretation of findings and drafting of the manuscript. All authors approved the final version of the manuscript.

**Funding** This work was supported by a Wellcome Trust Capacity Strengthening Strategic Award to the Public Health Foundation of India and a consortium of the UK universities. This research was part of Anamika Pandey's doctoral study at the London School of Hygiene and Tropical Medicine.

**Competing interests** None declared.

**Ethics approval** Institutional ethics committees of the Public Health Foundation of India and the London School of Hygiene and Tropical Medicine.

**Provenance and peer review** Not commissioned; externally peer reviewed.

**Data sharing statement** The authors confirm that all data underlying the findings are fully available without restriction. Data are publicly available and can be obtained from the Ministry of Statistics and Programme Implementation, Government of India, New Delhi: http://mospiold.nic.in/Mospi_New/site/inner.aspx?status=3&menu_id=37

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
