## [Reviewer comments · BMJ Open]

ARTICLE DETAILS

TITLE (PROVISIONAL)	Hospitalization trends in India from serial cross-sectional nationwide surveys: 1995 to 2014
AUTHORS	Pandey, Anamika; Ploubidis, George; Clarke, Lynda; Dandona, Lalit

VERSION 1 - REVIEW

REVIEWER	Enno Nowossadeck Robert Koch Institute Berlin, Germany
REVIEW RETURNED	24-Oct-2016

GENERAL COMMENTS	This paper analyses hospitalisation trends in India over a period of 20 years. In India a central registry of hospitalisations does not exist. Therefore, the authors use data from serial nationwide healthcare surveys. In the analytic part of the paper the authors analyse the factors contributing to the change in hospitalisation rates by using multivariate analyses and a regression decomposition technique. Major remarks: At the end of the introduction section the authors describe their intention but do not put a research question. The scientific purpose is not clear. That seems therefore problematic that the authors present a lot of data but cannot focus on main results of their analysis. In the measures section (not in the introduction section) the authors refer to the Anderson's model, but they don't use it in the discussion. A model based approach would be helpful to discuss the results. Minor remarks: Table 4-6: • Please add a measure of goodness of fit. Table 6: • Variable age: What is the reference category "young old"? What does it mean that, after adjusting for the covariates, age and place of residence ceased to be significant predictors of hospitalisation, meanwhile the need variables remain significant factors?
---

REVIEWER	Manisha Nair University of Oxford, UK
REVIEW RETURNED	23-Dec-2016

GENERAL COMMENTS	This paper examines the trends and reasons for hospitalisation in India using NSSO data with a particular focus on the differences
--

between two broad age groups <60 and ≥60 years. It also looks at the factors associated with hospitalisation in the ≥60 group. My comments are listed below:

Major comments

The justification that the population in India is rapidly ageing for looking at hospitalisation above 60 years is not correct. The current population pyramid of India shows that a majority are young or in early adulthood with projections of an ageing population only after 2050 (<http://www.worldlifeexpectancy.com/india-population-pyramid>).

It would be more useful to see how the trends in hospitalisation has changed for young adults over the years. Premature deaths are what a country should focus on averting. Can the authors categorise the population into groups of 10 years or at least <40, 41-50, 51-60, and >60 to examine the trends and reasons for hospital admission (Table S2 looks more useful). A finding that more hospital admissions occur in ≥60 years group compared to <60 seems somewhat intuitive considering life-expectancy in India. The socioeconomic disparities and public vs private analysis is fine although not a novel finding as the main health expenditure is out-of-pocket.

Methods

The authors do not comment on how any missing information on the determinants were handled. These surveys are prone to bias due to missing data. Please report the proportion of missing information for age, diseases categories and socioeconomic data.

Did the authors use any statistical method to account for the two-stage stratified design and possible clustering of sample?

Results

One of the important findings is the increase in contribution of NCDs to hospitalisation in the <60 years group (increased 2 times). This needs more discussion.

Considering the current population demography of India, the absolute numbers of hospital admission will be much higher in the <60 group, which would be more relevant for the purposes of planning resources for universal health coverage. Example a 2% increase in hospitalisation among <60 years would be far higher number of episodes in this group compared with a 2% increase in the older group. It is important to consider this aspect.

Discussion

I am not sure how the authors reached the following conclusion from their analysis (page 25), "We found that economic vulnerability hinders healthcare utilisation at all ages, but more so at older ages."

Minor comments

Introduction

Lines 7-14 "The difference between life expectancy and healthy life expectancy was 7.2 years for the male population and 8.0 years for the female population in 1990, which increased to 7.6 years and 9.4 years, respectively in 2013, suggesting that India's population loses more years of healthy life to disability today than it did 20 years ago" – from the differences (0.4 for females and 1.4 for males), it cannot be concluded that India's population now loses more years. I would say that it is stagnant or a slight increase.

	Results Along with the confidence intervals, it will be useful to see the p-values for χ^2 test for difference in proportion/ any other test that the authors used to assess difference in proportion across the groups.
--	--

VERSION 1 – AUTHOR RESPONSE

Reviewer: 1

Reviewer Name: Enno Nowossadeck

Institution and Country: Robert Koch Institute Berlin, Germany Please state any competing interests or state 'None declared': None declared

Please leave your comments for the authors below. This paper analyses hospitalisation trends in India over a period of 20 years. In India a central registry of hospitalisations does not exist. Therefore, the authors use data from serial nationwide healthcare surveys.

In the analytic part of the paper the authors analyse the factors contributing to the change in hospitalisation rates by using multivariate analyses and a regression decomposition technique.

Major remarks:

1. At the end of the introduction section the authors describe their intention but do not put a research question. The scientific purpose is not clear. That seems therefore problematic that the authors present a lot of data but cannot focus on main results of their analysis.

Reply: We have now clearly stated the two objectives of our paper on page 4 last paragraph. The key findings from our study have now been stated on page 28 paragraph 1 and we have focused on discussing these results in the subsequent paragraphs in the discussion section.

2. In the measures section (not in the introduction section) the authors refer to the Anderson's model, but they don't use it in the discussion. A model based approach would be helpful to discuss the results.

Reply: As suggested, we have now referred to the Andersen model in the discussion on page 31 last paragraph and page 33 paragraph 1. We have mentioned the Andersen model in the method section of our paper as this has been used to identify the independent variables relevant to attain our study objectives.

Minor remarks:

3. Table 4-6: Please add a measure of goodness of fit.

Reply: We have now added a measure of goodness of fit for the logistic model and the p-value for the Wald for the difference in coefficients in Table 6. In Table 5 we have now reported the 95% confidence intervals for the estimates, and the confidence intervals have already been included in Table 4. We believe that 95% confidence intervals are sufficient for assessing the difference in estimates across groups or overtime in Tables 4 and 5.

4. Table 6: Variable age: What is the reference category "young old"? What does it mean that, after adjusting for the covariates, age and place of residence ceased to be significant predictors of hospitalisation, meanwhile the need variables remain significant factors?

Reply: In Table 6, the reference category for age is 60-69 years. This sentence has now been re-

phrased on page 24 paragraph 1 to make it clearer.

Reviewer: 2

Reviewer Name: Manisha Nair

Institution and Country: University of Oxford, UK Please state any competing interests or state 'None declared': None declared

Please leave your comments for the authors below This paper examines the trends and reasons for hospitalisation in India using NSSO data with a particular focus on the differences between two broad age groups <60 and ≥60 years. It also looks at the factors associated with hospitalisation in the ≥60 group. My comments are listed below:

Major comments

1. The justification that the population in India is rapidly ageing for looking at hospitalisation above 60 years if not correct. The current population pyramid of India shows that a majority are young or in early adulthood with projections of an ageing population only after 2050 (<http://www.worldlifeexpectancy.com/india-population-pyramid>).

It would be more useful to see how the trends in hospitalisation have changed for young adults over the years. Premature deaths are what a country should focus on averting. Can the authors categorise the population into groups of 10 years or at least <40, 41-50, 51-60, and >60 to examine the trends and reasons for hospital admission (Table S2 looks more useful). A finding that more hospital admissions occur in ≥60 years group compared to <60 seems somewhat intuitive considering life-expectancy in India. The socioeconomic disparities and public vs private analysis is fine although not a novel finding as the main health expenditure is out-of-pocket.

Reply: As suggested, we now show the trends in hospitalisation rates for several age groups in Table 1. The age categorisation that we used is appropriate from the epidemiological view point in India. Although majority of India's population is still young or middle aged, it is important to have evidence on the changing demand for healthcare of older population for the planning of healthcare resources given the higher growth rate of older population than the overall population. The analyses of trends in socioeconomic disparities in hospitalisation and the use of public vs private hospitals across national sample surveys over 20 years in this paper are useful insights to inform further evolution of the health system.

Methods

2. The authors do not comment on how any missing information on the determinants were handled. These surveys are prone to bias due to missing data. Please report the proportion of missing information for age, diseases categories and socioeconomic data.

Reply: As suggested, we now show the proportion of missing data for the independent variables in Table S1. We also mention about handling the missing data in the method section on page 6 paragraph 1.

3. Did the authors use any statistical method to account for the two-stage stratified design and possible clustering of sample?

Reply: The statistical method that we used to account for sampling design of NSSO data has now been stated on page 10 last paragraph.

Results

4. One of the important findings is the increase in contribution of NCDs to hospitalisation in the <60

years group (increased 2 times). This needs more discussion.

Considering the current population demography of India, the absolute numbers of hospital admission will be much higher in the <60 group, which would be more relevant for the purposes of planning resources for universal health coverage. Example a 2% increase in hospitalisation among <60 years would be far higher number of episodes in this group compared with a 2% increase in the older group. It is important to consider this aspect.

Reply: We agree that this is a useful point to make. This is now discussed on page 31 paragraph 1.

Discussion

5. I am not sure how the authors reached the following conclusion from their analysis (page 25), “We found that economic vulnerability hinders healthcare utilisation at all ages, but more so at older ages.”

Reply: This has been rephrased on page 31 last paragraph to convey our findings in simple language.

Minor comments

6. Introduction

Lines 7-14 “The difference between life expectancy and healthy life expectancy was 7.2 years for the male population and 8.0 years for the female population in 1990, which increased to 7.6 years and 9.4 years, respectively in 2013, suggesting that India’s population loses more years of healthy life to disability today than it did 20 years ago” – from the differences (0.4 for females and 1.4 for males), it cannot be concluded that India’s population now loses more years. I would say that it is stagnant or a slight increase.

Reply: This sentence has now been re-phrased in the introduction on page 4 paragraph one.

Results

7. Along with the confidence intervals, it will be useful to see the p-values for χ^2 test for difference in proportion/ any other test that the authors used to assess difference in proportion across the groups.

Reply: We have now added the p-value for the t-test for the difference in coefficients in Table 6 and reported the 95% confidence intervals for the estimates in Table 5. The confidence intervals for the estimates have already been included in Table 4. We believe that 95% confidence intervals are sufficient for assessing the difference in estimates across groups or overtime in Tables 4 and 5.

VERSION 2 – REVIEW

REVIEWER	Enno Nowossadeck Robert Koch Institute Berlin Germany
REVIEW RETURNED	20-Feb-2017

GENERAL COMMENTS	I’d like to thank the authors for revision of their paper. It’s now much more comprehensible. So I only have some minor comments. • The authors now stated their objectives in the last paragraph of the introduction section. But they have failed to amend the abstract accordingly. Please make up leeway.• The first paragraph on page 4 is capable of being misunderstood. The Global Burden of Disease Study 2013 (Murray et al. 2015, Ref. 2) quantified an increase in healthy life expectancy for women and men (9 and 6.4 years, respectively), even though the difference between life expectancy and healthy life expectancy increased in
---

	2013. Therefore data don't "suggesting that India's population continues to lose years of healthy life to disability".  • Table 4-7, table S1 and page 17, row 21: Please explain "SC/ST" for non-Indian readers, for example in the measures section. There is only a term "social group" (page 7, row 7). • Page 24, row 3: "health sector", please insert a "I" • The keyword "expansion of morbidity" seems to be problematic because this term does not play a substantial role in the paper. • Please add page numbers in Reference 2.
--	---

REVIEWER	Manisha Nair University of Oxford, UK
REVIEW RETURNED	16-Feb-2017

GENERAL COMMENTS	The authors have addressed a majority of the comments, but I have two points which need further consideration:  1. Adjustment for clustering effect: The authors mention that they used survey weights to adjust for the stratified design. However, this does not adjust for clustering which requires adjustment to the standard errors using design effect (if known) or other methods such as calculating Robust SE. 2. Missing data: The authors say that they dropped all observations with missing data. Instead, it might be better to say that they did a complete case analysis. Table S1 is a useful addition.
--

VERSION 2 – AUTHOR RESPONSE

Reviewer: 2

Reviewer Name: Manisha Nair

Institution and Country: University of Oxford, UK Please state any competing interests or state 'None declared': None declared

Please leave your comments for the authors below The authors have addressed a majority of the comments, but I have two points which need further consideration:

1. Adjustment for clustering effect: The authors mention that they used survey weights to adjust for the stratified design. However, this does not adjust for clustering which requires adjustment to the standard errors using design effect (if known) or other methods such as calculating Robust SE.

Reply: To calculate the 95% confidence interval we have taken the clustering effect into account. We have verified that estimated confidence intervals are wider than what they would have been with a simple random design assumption. This has now been clarified in the method section on page 8 last paragraph.

2. Missing data: The authors say that they dropped all observations with missing data. Instead, it might be better to say that they did a complete case analysis. Table S1 is a useful addition.

Reply: This has now been modified in the methods section on page 5 paragraph 2.

Reviewer: 1

Reviewer Name: Enno Nowossadeck

Institution and Country: Robert Koch Institute Berlin, Germany Please state any competing interests or state 'None declared': None declared

Please leave your comments for the authors below I'd like to thank the authors for revision of their paper. It's now much more comprehensible. So I only have some minor comments.

1. The authors now stated their objectives in the last paragraph of the introduction section. But they have failed to amend the abstract accordingly. Please make up leeway.

Reply: The objectives section in the abstract has been modified.

2. The first paragraph on page 4 is capable of being misunderstood. The Global Burden of Disease Study 2013 (Murray et al. 2015, Ref. 2) quantified an increase in healthy life expectancy for women and men (9 and 6.4 years, respectively), even though the difference between life expectancy and healthy life expectancy increased in 2013. Therefore data don't "suggesting that India's population continues to lose years of healthy life to disability".

Reply: This has been removed from the introduction.

3. Table 4-7, table S1 and page 17, row 21: Please explain "SC/ST" for non-Indian readers, for example in the measures section. There is only a term "social group" (page 7, row 7).

Reply: This has now been explained in Table 4-7, Table S1, and on page 7 as a footnote.

4. Page 24, row 3: "heath sector", please insert a "l"

Reply: This has now been rectified on page 24, row 3.

5. The keyword "expansion of morbidity" seems to be problematic because this term does not play a substantial role in the paper.

Reply: This has now been removed from the list of keywords.

6. Please add page numbers in Reference 2.

Reply: Volume, issue and page numbers have now been added in Reference 2.

VERSION 3 – REVIEW

REVIEWER	Enno Nowossadeck Robert Koch Institute Berlin
REVIEW RETURNED	05-Apr-2017

GENERAL COMMENTS	I don't have further comments.
--------------------------------

REVIEWER	Manisha Nair University of Oxford, UK
REVIEW RETURNED	03-Apr-2017

GENERAL COMMENTS	The authors have addressed the comments, but I would have liked the authors to explicitly state the method that they used to adjust for the clustering effect instead of simply writing that the 95% CI have been adjusted to account for clustering.
---

VERSION 3 – AUTHOR RESPONSE

Reviewer: 2

Reviewer Name: Manisha Nair

Institution and Country: University of Oxford, UK

Please state any competing interests or state 'None declared': None declared

Please leave your comments for the authors below

The authors have addressed the comments, but I would have liked the authors to explicitly state the method that they used to adjust for the clustering effect instead of simply writing that the 95% CI have been adjusted to account for clustering.

Reply: We have now mentioned the method used to adjust for the survey design features of NSSO data on page 8 last paragraph.